# Antioxidant Behavior of Carbon/Carbon Composites with Hot Dip Plating and Electroplating for Single-Crystal Furnaces

**DOI:** 10.3390/ma17235798

**Published:** 2024-11-26

**Authors:** Zuxing Qi, Chaofan Du, Guoying Bao, Shan Wang, Dedong Gao, Haixing Lin, Yan An

**Affiliations:** 1Department of Mechanical Engineering, Qinghai University, Xining 810016, China; ys220805030205@qhu.edu.cn (Z.Q.); ys210805010157@ghu.edu.cn (C.D.); ys230805020201@qhu.edu.cn (G.B.); gaodd@qhu.edu.cn (D.G.); ys220802030171@qhu.edu.cn (H.L.); 2Sichuan Gokin Solar Technology Co., Ltd., Yibin 644600, China; yan.an@gokinsolar.com

**Keywords:** carbon/carbon composites, composite coating, hot-dip plating, antioxidation performance

## Abstract

In the Czochralski single-crystal silicon manufacturing industry, single-crystal furnaces often experience corrosion from silicon vapor, which reduces their operational lifespan. However, the preparation of metal coatings on the surface of C/C composites is challenging due to their low coefficient of thermal expansion and the intricate structure of carbon fibers. To address this issue and achieve high-quality alloy coatings, Ni-Al and Ni-Al/Si composite coatings are successfully prepared on the surface of C/C composites through a combination of electroplating and hot-dip plating, and their oxidation behavior at elevated temperatures is thoroughly investigated. The experimental results indicate that the Ni-Al composite coatings exhibit superior antioxidant properties compared to Ni coatings following thermal shock experiments, thereby significantly enhancing the antioxidant performance of C/C composites. This improvement is attributed to the preferential oxidation of surface aluminum, which forms a dense Al_2_O_3_ layer in aerobic and high-temperature environments, effectively preventing oxygen from reaching the underlying matrix. During the oxidation process, coating elements migrate outward along the concentration gradient, while oxygen molecules diffuse inward. Simultaneously, aluminum atoms diffuse inward, and Ni atoms diffuse outward, where they partially dissolve with oxygen. The inner coating’s Ni enhances the bonding of the coating by connecting the substrate to the outer layer. Meanwhile, the added Si in the Ni-Al/Si composite coating further improves the antioxidant properties of the coating.

## 1. Introduction

Silicon solar cells are moving towards higher efficiency and lower cost. Cast-type single-crystal silicon is a promising material for achieving this goal [1]. In recent years, due to the growing demand for thermal field stability and improved growth quality in single-crystal furnaces, Carbon/Carbon (C/C) composites have been extensively utilized in the production of single-crystal furnaces. As a kind of high-temperature-resistant and high-strength material, C/C composites have attracted much attention in the field of aerospace and crystal manufacturing. In particular, the application of C/C composite materials has emerged as a predominant trend in the high-temperature regions of single-crystal furnaces, including components such as the crucible, flow guide tube, and upper and lower insulation tubes. As integral components of the single-crystal furnace, these elements directly influence the stability of the thermal field, as well as the quality and growth rate of monocrystalline silicon. Fu et al. consolidate recent advancements in micro/nano multiscale strategies—such as nanoparticles, nanowires, and graphene—to enhance high-temperature oxidation and ablation resistance in C/C composites. Their paper concludes with challenges and future directions for achieving superior mechanical and thermal performance, aiming to inspire scientific and industrial advancements in robust C/C materials [2,3,4]. However, the broader application of C/C composites is restricted by their limitations in temperature oxidation resistance. It has been found that C/C composites are oxidized at 450 °C and burned at 550 °C in an aerobic environment. Han et al. investigate the impact of oxidation on residual mechanical properties for safe in-service use. They propose a microscale degradation model based on mass conservation and diffusion to predict oxidation behavior and use simulations to show a 24–32% strength reduction after 30 min at 850 °C. Their findings reveal that oxidation redistributes stress, weakens load capacity, and increases oxygen diffusion in stressed areas of the pyrocarbon matrix [5,6,7]. C/C composites are primarily composed of carbon fiber mesh and pyrolytic carbon powder. The carbon mesh planes are typically characterized by unsaturated chemical bonds and free-electron fringe carbon atoms, which are regarded as active adsorption sites. These sites are prone to adsorption of surrounding oxidizing gasses, and oxidation reactions occur at temperatures of about 400 ° C. During oxidation, the initial erosion occurs at the material’s surface, including pores and the fiber/matrix interface. This is followed by the erosion of the anisotropic and isotropic matrix carbon, then the fiber side surfaces extending to their ends, and ultimately the core of the fibers. Oxidative corrosion initially targets the matrix carbon with a low degree of graphitization, which is progressively removed as corrosion causes pore expansion. Concurrently, as the temperature rises, the carbon fibers experience gradual radial erosion [8,9].

To enhance the oxidation resistance of C/C composites, two primary methods are employed: substrate modification and coating preparation. Coating preparation involves applying a protective layer that prevents oxygen infiltration without altering the substrate’s inherent properties. The antioxidant design idea of the coating is shown in Figure 1 below, which has the advantages of being simple, convenient, and cheap [10,11]. Composite coatings have been extensively studied by researchers worldwide for their superior antioxidant properties. Zhu et al. successfully prepared SiC-ZrSi_2_ gradient composite coatings on the surface of C/C composites using the embedding and slurry methods. The results demonstrated that the coatings exhibited optimal antioxidant properties at 1200 °C, 1300 °C, and 1500 °C, and showed improved oxidation resistance at medium and high temperatures [12]. Li et al. investigated the protective effect of a three-layer gradient self-healing coating on C/C composites, consisting of an inner layer of B_4_C and β-SiC, a middle layer of SiC, and an outer sealing layer of SiO_2_. The results indicated that the coating exhibited effective self-healing at 1500 °C, with an oxidative weight change rate of 1.3% after 50 h at 1500 °C, along with excellent thermal shock resistance [13]. Huang et al. synthesized Al-doped SiC coatings on C/C composites using the filler cementation technique to study the microstructure, composition, and antioxidant properties of the coatings at different preparation temperatures (1500~2100 °C). The results showed that coatings prepared at 1500~1900 °C were discontinuous 3C-SiC/2H-SiC/C/Si/Al₂O₃ coatings with some pores and cracks. In contrast, coatings prepared at 2100 °C consisted of 4H-SiC with dissolved Al atoms and exhibited fewer cracks. The coatings obtained at 2100 °C demonstrated the best antioxidant properties due to the in situ formation of Al_2_O_3_ and SiO_2_ oxide layers on the surface [14]. Chen et al. investigate Cr-coated Zr alloys as promising ATF cladding materials. In their study, they apply Cr coatings to 1400 mm N36 tubes using an industrial arc system. Orthogonal analyses identify key process parameters affecting coating characteristics, including surface roughness, defects, and crystal orientation [15]. In summary, compared to single coatings, composite coatings are more effective at mitigating or preventing high-temperature oxidation reactions, thereby reducing performance degradation and extending the lifespan of C/C composites. These coatings are crucial for enhancing the antioxidant properties of C/C composites [16].

Hot-dip plating, a straightforward method for preparing dense coatings, is frequently employed in coating metal substrates [17]. Hot-dip plating not only significantly enhances the corrosion resistance and high-temperature oxidation resistance of the substrate material but also improves the material’s surface appearance and imparts a certain level of comprehensive mechanical properties [18,19]. The hot-dip plating process, currently encompassing two primary systems, involves the first method known as the protective gas method [20,21,22]. This technique employs a redox process to remove oil and oxides from the substrate surface, after which the substrate is introduced into a molten aluminum or aluminum alloy bath within a reducing atmosphere. The second system is the flux-assisted hot-dip plating method. This process entails removing contaminants and oxides from the substrate’s surface, immersing the substrate in a flux solution to form a coating aid film, and finally submerging the treated part into molten aluminum to complete the aluminum plating process.

The use of auxiliary plating agents can activate the metal surface, improve the infiltration of aluminum to the steel matrix, and play an important role in protecting the surface of the plating from oxidation or contamination. In the molten aluminum solution, the film of the plating aid dissolves, allowing the exposed steel surface to come into direct contact with the aluminum. This promotes rapid wetting of the surface, leading to the formation of a continuous and uniform aluminum coating [23]. Cheng studied the microstructure evolution of intermetallic layers of hot-dip-aluminized low-carbon steel with silicon added [24]. The thickness of the coating decreased with the increase in Si content. The metal alloy layer consists of a FeAl_3_ layer with an outer layer thickness of 5 μm and the main Fe_2_Al_5_ layer in the inner layer, which is mainly because silicon occupies the C-axis vacancy on Fe_2_Al_5_. The diffusion rate of iron and aluminum in the Fe_2_Al_5_ layer was slowed down. Zhang et al. employed hot-dip plating to deposit a WSi_2_ coating on a tungsten substrate [25]. The results indicated that the tungsten–silicide coating consists of a WSi_2_ layer and an interfacial W_5_Si_3_ layer, with the silicon concentration gradually decreasing and the tungsten concentration correspondingly increasing along the diffusion direction. The hot-dip plating temperature has a significant effect on the coating thickness and grain size, and the prepared coating has strong oxidation resistance. The combination of electroplating and melt hot-dip plating is applied to prepare coatings on C/C composites. This advances hot-dip plating techniques in C/C composite coatings and provides a novel approach for future development in the field.

Currently, C/C composite coating preparation methods include magnetic sputtering, chemical vapor deposition (CVD), the sol–gel method, and plasma spraying [10,26,27]. These techniques are characterized by complex processes, high costs, and limited applicability to small-scale components [28]. Meanwhile, rare-earth metals are often added to the coating materials to improve the antioxidant properties of the coatings, which further increases the cost [29,30]. In recent years, researchers have dedicated efforts to exploring novel coating preparation techniques, such as plasma-enhanced chemical vapor deposition (PCVD) and molten salt deposition, with the aim of addressing current challenges in the coating process, reducing production costs, and improving coating performance [31]. In addition, it is also crucial to continuously innovate the traditional coating preparation technology and explore new raw materials and processes, such as the development of new glass coatings, precious metal coatings, ceramic coatings, and so on [32].

Considering the challenges associated with depositing metal coatings on C/C composites, which arise from their low thermal expansion coefficient and intricate carbon fiber structure, a novel method is proposed for the preparation of Ni-Al and Ni-Al/Si composite coatings on C/C composite substrates, involving Ni electroplating followed by Al hot-dip plating. This method reduces corrosion of the boiler by silicon vapor in the Czochralski single-crystal silicon manufacturing industry.

## 2. Experimental Materials and Methods

### 2.1. Experimental Methods

The detailed experimental procedure is illustrated in Figure 2, which comprehensively represents the steps involved. The C/C composite material is processed as 15 mm × 15 mm × 5 mm. A Ni (Shanghai Macklin Biochemical Technology Co., Ltd., Shanghai, China) coating with a thickness of about 13 µm is deposited on the outer surface of the C/C composites by electroplating. Al (Shanghai Macklin Biochemical Technology Co., Ltd.) and Al-Si coatings are then applied on the Ni layer using hot-dip plating. Finally, Ni-Al and Ni-Al/Si composite coatings are formed.

The C/C composites are polished with 400#, 600#, and 800# grit sandpaper prior to plating, resulting in a clean and smooth cut. Subsequently, ultrasonic cleaning is conducted with deionized water and anhydrous ethanol in succession, followed by drying in an oven at 130 °C for 3 h and storage in a desiccator. The plating solution requires temperature control during the plating process, maintaining the solution temperature at 55 °C, the pH between 5.8 and 6.2, the current density at 0.4 A/dm^2^, the magnetic stirring speed at 500 rpm, and the plating time at 40 min.

The components of the plating solution for Ni deposition on the surface of C/C composites are listed in Table 1.

The Ni-coated samples are cleaned using ultrasonic bubbles in acetone and anhydrous alcohol, followed by low-temperature drying to remove surface oil and chemical residues. The sample intended for hot dipping requires pre-treatment with an auxiliary plating process. The composition of the plating aid is shown in Table 2.

The auxiliary plating is performed at 87 °C for 10 min, after which Al and Al/Si hot-dip coatings are applied under inert gas protection. Finally, Ni-Al and Ni-Al/Si composite coatings are successfully fabricated. The addition of silicon improves both the wear resistance and the corrosion resistance of the material by reducing the coating thickness [22]. Due to the significantly higher melting point of silicon compared to aluminum, Al/Si ingots are utilized to introduce silicon into the aluminum for easier melting. The effects of temperature, holding time, and silicon content on the antioxidant properties of the coating are investigated through three-factor, three-level orthogonal experiments. The corresponding parameters are provided in Table 3.

### 2.2. Coating Characterization

The surface and cross-sectional morphologies of the coatings before and after oxidation are observed using a JEOL (JSM-7900F, JEOL Ltd., Tokyo, Japan) field emission electron microscope. The field FESEM offers high resolution, with secondary electron images reaching 0.7 nm at 15 kV. It has a magnification range from 25× to 1,000,000× and an acceleration voltage range of 0.01 kV to 30 kV. The microscope features a Schottky field emission electron gun with an electron beam current range of 1 pA to 500 nA. The working distance is automatically adjustable from 2 mm to 40 mm, with manual adjustment within 2 mm. The optimal working distance for EDS is 10 mm. An EDS is used for the quantitative analysis of the elemental composition in the microregions of the coating. Additionally, a D/max2500PC X-ray diffractometer (Rigaku, Tokyo, Japan) is used to characterize the phases of the prepared coatings and the oxide phases formed after high-temperature oxidation. The XRD (Rigaku, Tokyo, Japan) uses a copper (Cu) target as the anode, with a scintillation counter. Test mode settings include a step size of 0.02°and a speed of 8° per minute, using a 2-Ө/Ө linkage. Data analysis is conducted using JADE 9 (MDI-R98218) software.

### 2.3. High-Temperature Oxidation Test

A thermal shock test is conducted to evaluate the oxidation resistance of the coating [33]. Thermal shock tests are conducted in a box-type resistance furnace at temperatures of 450 °C, 500 °C, 550 °C, 600 °C, 650 °C, and 700 °C. When the furnace temperature reaches the specified level, the sample is placed in the furnace and held at that temperature for 30 min, followed by cooling at room temperature for 5 min. This constitutes 1 cycle, with a total of 20 cycles performed. The weight change rate of the thermal shock experimental coating at intermediate temperatures is used as the criterion for evaluating the results of the orthogonal experiment. Subsequently, the thermal shock experiment is conducted at temperatures ranging from 800 °C to 1400 °C for the C/C composite materials, Ni electroplated samples, and the optimal hot-dip plating solution, following the aforementioned experimental procedures. The macro-morphology is observed in the cycle and the weight change rate is calculated. The formula for calculating its oxidized weight rate is as follows:$$\text{∆}W=(M_{2}-M_{1})/M_{1}$$
where Δ*W* is oxidized weight change in the specimen (%), M_2_ is mass after oxidation (g), and M_1_ is mass before oxidation (g).

### 2.4. Silicon Vapor Corrosion Test

Silicon vapor corrosion tests are conducted on the prepared Ni coating samples, composite coating samples, and C/C composite materials. The experimental process is as follows:Sample pretreatment: The coated samples and C/C composites materials are subjected to degreasing and decontamination procedures to eliminate any impurity elements that could potentially influence the corrosion results during the experimental process.Corrosion experiment: The sample is positioned in a crucible with a layer of monocrystalline silicon raw material added to the bottom. The crucible is then subjected to multiple vacuum cycles, followed by the introduction of argon to purge the furnace of air. The temperature is subsequently raised to 1450 °C and maintained for 5 h. After cooling, the sample is extracted and analyzed using microscopic morphology observation and energy-dispersive spectroscopy.

## 3. Results and Analysis

### 3.1. Analysis of Orthogonal Experiment Results

The effects of temperature, time, and silicon content on the oxidation resistance of the coating are evaluated through orthogonal experiments. The range R calculation method can be employed to ascertain the influence ranking of each factor on the coating quality, while mean value calculations can be used to identify the optimal parameter combination. The antioxidant performance of the coatings is used as a criterion, and the results are presented in Table 4 below. The order of influence on antioxidant performance, determined through the calculation of the extreme deviation R, is as follows: B > A > C. The optimal parameter combination is presented in Table 4 below. Based on the mean value calculation, the optimal parameters are A_1_B_2_C_2_, which correspond to a time of 60 min, a temperature of 740 °C, and a silicon content of 2%. Temperature is the most critical factor affecting antioxidant properties, followed by time, while silicon content has a relatively minor impact. This indicates that, in improving the antioxidant properties of coatings, priority should be given to controlling the temperature, with appropriate extension of the treatment time. The limited effect of silicon content on antioxidant performance suggests that variations in silicon content have a minimal impact on the antioxidant properties of the coatings within the studied range. Details are in Appendix A.

The nine groups of samples undergo thermal shock experiments to investigate the optimal hot-dip plating parameters and the influence of each parameter. The results are presented in Figure 3. The figure shows that the weight change rate of the specimens increases with temperature during the thermal shock experiment. This increase is attributed to the accelerated oxidation reaction at higher temperatures, which facilitates greater oxygen diffusion into the coating. However, the weight change rate decreases significantly when the temperature exceeds 600 °C. This decrease is explained by the oxidation of the Al element on the surface and the formation of a protective Al_2_O_3_ layer, which impedes further oxygen penetration. The weight change observed in the specimens is primarily influenced by oxidizing reactions occurring at high temperatures. As the temperature rises, thermal energy accelerates chemical reactions, enhancing the rate of oxidation. Consequently, more oxygen atoms react with the coating material, leading to increased weight change. Additionally, the diffusion rate of oxygen through the coating is also affected by the oxidation temperature; higher temperatures facilitate faster oxygen diffusion, further contributing to the increased weight change. The weight change rate decreases above 600 °C, which can be attributed to the formation of a dense Al_2_O_3_ layer on the coating’s surface. This Al_2_O_3_ layer acts as a barrier, preventing further oxygen penetration. It is formed through the oxidation of the Al element present on the coating’s surface. This layer effectively restricts the interaction between oxygen and the underlying coating material, thereby reducing the rate of weight change.

In conclusion, the thermal shock experiments demonstrated that the weight change rate of the samples increased with rising temperature due to the enhanced oxidation reactions. Beyond the critical temperature of 600 °C, the weight change rate decreased, primarily attributed to the formation of a protective Al_2_O_3_ layer that impedes further oxygen entry.

### 3.2. XRD Scanning Results After Coating and Coating Oxidation

As shown in Figure 4, the X-ray diffraction patterns of the oxidized composite coating, the hot-dip aluminum composite coating, and the electroplated Ni coating are presented in Figure 4a–c. After comparison with the PDF (98-000-0062) standard card, in Figure 4a, the peaks of Al are mainly 38.474° (111), 44.723° (200), 65.1° (220), 78.234° (311), and 82.441° (222). After comparison with the PDF (04-002-3621) standard card, in Figure 4a, the peak of Al_2_O_3_ is 35.133° (104). After comparison with the PDF (98-000-0062) standard card, in Figure 4b, the peaks of Al are mainly 38.474° (111), 44.723° (200), 65.1° (220), 78.234° (311), and 82.441° (222). After comparison with the PDF (04-002-8298) standard card, in Figure 4c, the peaks of Ni are 41.785° (002) and 58.532° (102). The absence of Ni and C diffraction peaks in the spectrum of the hot-dip aluminum plating indicates that an Al coating has been successfully formed on the surface of the substrate. The diffraction pattern after oxidation reveals a decrease in Al content and an increase in Al_2_O_3_ content following the oxidation process. Additionally, based on the XRD angles, the alumina formed is identified as α-Al_2_O_3_. This type of alumina possesses advantages such as high hardness, a high melting point, excellent thermal conductivity, and strong corrosion resistance, providing effective protection for the substrate [34]. Moreover, Al remains present on the surface of the coating, and no diffraction peak for the C element is detected. In summary, the prepared coating effectively protects the substrate and enhances the oxidation resistance of the coating.

The X-ray diffraction patterns lead to the following conclusions: The absence of C-element diffraction peaks in the hot-dip Al composite coating confirms the successful coverage of the C/C composite substrate by the coatings. Overall, the prepared coatings effectively protect the substrate and enhance its antioxidant properties.

### 3.3. Coating SEM Scanning Results and Microstructure

Figure 5 shows the microscopic morphology of the coating surface after the experiment. The Ni and Al coatings, formed through electroplating and hot-dip coating treatments, are successfully deposited on the surface of the C/C composites and effectively cover both the carbon fibers and the pyrolytic carbon substrate. The C/C composites mainly consist of carbon fibers and pyrolytic carbon as indicated in Figure 5a, and the surface of the C/C composites is not homogeneous with the simultaneous existence of pyrolytic carbon particles, so it is necessary to polish the substrate smooth before preparing for the coating process. In Figure 5b,c, the Ni coating exhibits a few surface holes, but the overall coverage is adequate. The C/C composites can produce hole defects during oxidation due to the thermal exposure area and oxidation time [35]. Adsorption occurs on the surface of the substrate despite the presence of defects. In contrast, the surface of the hot-dip aluminum coating is uniform and smooth, with no carbon fibers or other substrate materials visible. These coatings can effectively protect the substrate materials and improve their antioxidant properties. In contrast, the hot-dip Al/Si coating shown in Figure 5d has a rough surface. While pores and cracks are present, they are confined to the surface and do not cause immediate failure of the coating. The thermal shock experiments further demonstrate that the coating containing silicon exhibits better high-temperature oxidation resistance.

### 3.4. Coating EDS Analysis

The experimental results of the C/C composite surface coating preparation are thoroughly observed and analyzed, and the EDS energy spectra of the three coating surfaces are obtained through scanning electron microscopy. In the first set of experiments, the Ni coating is successfully prepared, as shown in Figure 6a, exhibiting a stable elemental composition primarily consisting of Ni, C, and O. The Ni coating is primarily derived from the C/C composite matrix. The carbon element originates mainly from the C/C composite substrate, while the oxygen element may be introduced from multiple sources, such as dissolved oxygen in the plating solution, oxygen exposure and adsorption from the air after electroplating, or oxygen introduced during sample preparation. The appearance of other element peaks mainly comes from impurities brought by sample preparation and instrument residues. Combined with the XRD analysis results, it can be seen that O only appears on the surface, and it has a relatively small content. It can be concluded that the Ni coating prepared in this experiment can be successfully deposited on the surface of C/C composites and has a stable element composition.

In the second set of experiments, the Al coating is successfully prepared, as shown in Figure 6b, with an elemental composition primarily consisting of C, O, and Al. The mass fraction of Al is 39.7%, which is lower than that of Ni in the Ni coating, making the Al coating relatively more fragile. However, the Al coating still has good encapsulation and protection for the C/C composite substrate. It is important to note that the contents of C and O elements are 42.2% and 12.5%, respectively, primarily originating from the C/C composite substrate and various introduced factors. Other elements detected in the energy spectra are mostly impurities introduced during sample preparation or instrument residues, but their relative contents are minimal.

In the third set of experiments, Al/Si coatings are successfully prepared, as shown in Figure 6c. Their elemental composition primarily consists of C, O, Al, and Si, along with trace amounts of Ca, K, Ag, and other elements. The mass fraction of C is 36.6%, which is lower than the previous two experiments, while the mass fraction of Al is 48.8%, which is higher than the previous two experiments. The Si content in the Al/Si coatings meets the established requirement of 2%. The presence of Si contributes to increased hardness and wear resistance of the Al/Si coatings compared to the other two coatings, although it also makes the preparation process more challenging [36]. The energy spectra indicate the elements of Al/Si contained in the prepared coatings, and the Al/Si coatings are successfully prepared on the surface of the C/C composites. It should be noted that in all three coatings, the elements C and O have high contents, mainly from the C/C composite substrate as well as a variety of introduced factors. The energy spectra reveal other elements that are primarily impurities introduced during sample preparation and residues from the instrumentation, with their relative concentrations being minimal.

In summary, three coatings were successfully prepared in this study, and information on their elemental compositions and inclusions was obtained by SEM observation and EDS spectrogram analysis.

### 3.5. High-Temperature Oxidation Performance of Coatings

This experiment investigates the thermal shock properties of the C/C composite, electroplated Ni, and hot-dip Ni/Al composite coatings. The experimental results show that, as shown in Figure 7, the oxidation weight change in C/C composites gradually increases with the increase in temperature. Above 650 °C, the oxidation weight change accelerates, peaking at 800 °C. Subsequently, the rate of weight change exhibits undulating fluctuations within the temperature range of 800~1200 °C. Beyond 1200 °C, the rate of oxidation weight change decreases due to extensive oxidation of the C/C composites and the very limited number of remaining specimens.

For the electroplated Ni-coated specimens, the weight change rate increases sharply once the temperature exceeds 500 °C. This increase is attributed to the oxidizing reaction between the Ni in the coating and the air, with some of the oxidized material escaping from the specimen’s surface, thereby increasing the weight change rate. During the oxidation process, the weight change rate decreases due to the presence of Ni oxide. After exceeding 900 °C, the coating starts to fail and the oxidation efficiency rises sharply. The oxidation weight change rate of the hot-dip aluminum and Al/Si alloy composite coatings remained relatively stable. However, as the temperature increased, the weight change rate of these coatings began to decrease. This decrease is attributed to the formation of Al_2_O_3_ on the surface of the coatings, which inhibits further oxidation. This indicates that the composite coatings effectively enhance the antioxidant performance of C/C composites and exhibit better antioxidant properties compared to the individual Ni coatings. They are still able to maintain a low weight change at 1400 °C.

### 3.6. Micro-Morphology of Coatings After Oxidation

Figure 8 shows the SEM microscopic morphology of the experimental specimens after the thermal shock resistance experiment. As shown in Figure 8, the coating adheres to the substrate materials. The SEM observations indicate that the unoxidized electroplated Ni coatings in Figure 8a exhibit a laminated structure. This structure provides effective protection to the substrate materials. In Figure 8b, after the oxidation experiment, an oxide layer forms on the surface of the Ni coating, which enhances its antioxidant performance. However, the Ni oxide exhibits poor adhesion to the substrate, resulting in coating delamination and a loss of protective effect on the substrate. For the samples treated with hot-dip aluminum plating (Figure 8c), the oxidation experiment results show that external oxygen preferentially reacts with the coating metal to form the α-Al_2_O_3_ oxide. This α-Al_2_O_3_ film acts as a barrier, preventing further oxygen erosion. Meanwhile, the internal aluminum remains, continuing to capture free oxygen and thereby maintaining the integrity of the substrate material. In Figure 8d, the layer thickness after the electroplating and hot-dipping methods is 36.62 μm. In summary, the preparation of this composite coating successfully enhances the antioxidant properties of C/C composites.

### 3.7. Silicon Vapor Corrosion Test Results for Coatings

The images obtained under the scanning electron microscope, shown below, reveal the results of silicon vapor etching conducted on C/C composites, electroplated Ni specimens, and hot-dip-plated aluminum specimens. The specific procedure for the silicon vapor etching experiment involved first cleaning the specimen’s surface with anhydrous ethanol to remove oil and impurities. The specimen, along with monocrystalline silicon, was then placed in a tube furnace and held at 1450 °C for 5 h. Figure 9a,d demonstrate that, after silicon vapor etching, a significant amount of Si is deposited on the surface of the carbon fibers in the C/C composites, altering the mechanical properties of the materials. Ultimate tensile strength increases with increasing Si content in Al/Si alloys [36]. In contrast, Figure 9b,e show that the C/C composites with Ni coatings exhibit minimal Si deposition on the surface after corrosion. Due to the protection provided by the Ni coatings, Si deposition does not come into contact with the carbon fibers, thereby preserving the mechanical properties of the materials. Meanwhile, Figure 9c,f reveal that the aluminum coatings do not adsorb Si deposition, and the composite coatings formed at high temperatures effectively protect the matrix. In the EDS analysis, Si is not tested in the mapping pattern due to its low content. Overall, the composite coating treatment of C/C composites enhances their antioxidant properties and improves their resistance to silicon vapor corrosion, thus increasing their potential for application in single-crystal furnaces.

## 4. Conclusions

This study aims to develop composite coatings with superior resistance to high-temperature oxidation on the surface of C/C composites using a combination of electroplating and hot-dip plating methods. Ni coatings are successfully applied to the surface of C/C composites using the electroplating method and are shown to effectively enhance the antioxidant properties of the substrate. Subsequently, Ni-Al and Ni-Al/Si composite coatings are successfully prepared through hot-dip plating. Microscopic examination reveals good bonding between these coatings and the substrate materials. It is also observed that the Ni coatings are depleted during the hot-dip plating process. Orthogonal experiments are conducted to investigate the effects of temperature, holding time, and silicon content on the oxidation resistance of the coating. The results reveal that the influence on the oxidation resistance of the coating follows the order temperature > holding time > silicon content. Additionally, the optimal parameters for coating preparation are identified as A_1_B_2_C_2_, which corresponds to a time of 60 min, a temperature of 740 °C, and a silicon content of 2%. The thermal shock test results demonstrate that all three coatings significantly enhance the oxidation resistance of the substrate, with the Ni-Al/Si composite coating exhibiting the best performance. This improvement is attributed to the formation of a dense α-Al_2_O_3_ film on the coating’s surface during oxidation, which effectively blocks oxygen penetration. Additionally, the inclusion of Si elements fills oxidation gaps and further prevents oxygen immersion. In contrast, the oxidation performance of the Ni-Al composite coating is reduced due to the presence of a small number of voids that develop during the oxidation process. The hot-dip plating technology has been successfully applied to the surface coating preparation of C/C composite materials, offering a novel approach in this field. Future research should explore the effects of different hot-dip plating aids, materials, and other factors on coating properties to further optimize their performance and application. The findings have both academic and practical significance, contributing to the enhancement of C/C composites’ quality and the acceleration of their industrial applications.

## Figures and Tables

**Figure 1 materials-17-05798-f001:**
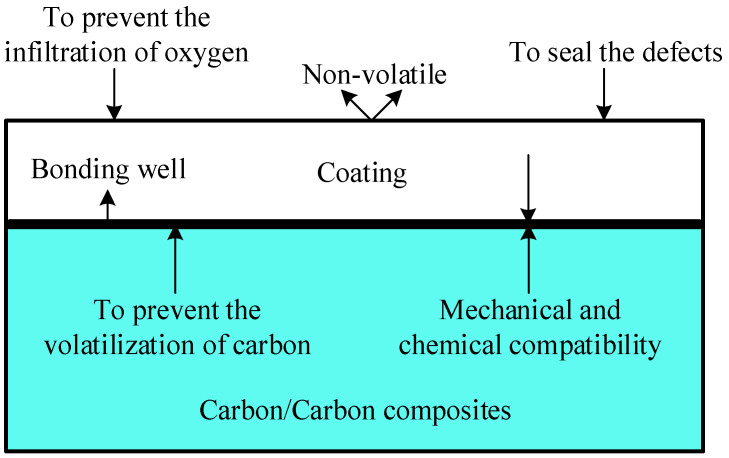
Antioxidant coating design conditions.

**Figure 2 materials-17-05798-f002:**
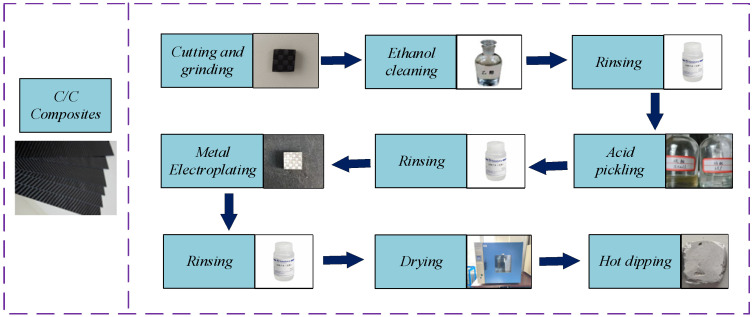
Experimental flowchart.

**Figure 3 materials-17-05798-f003:**
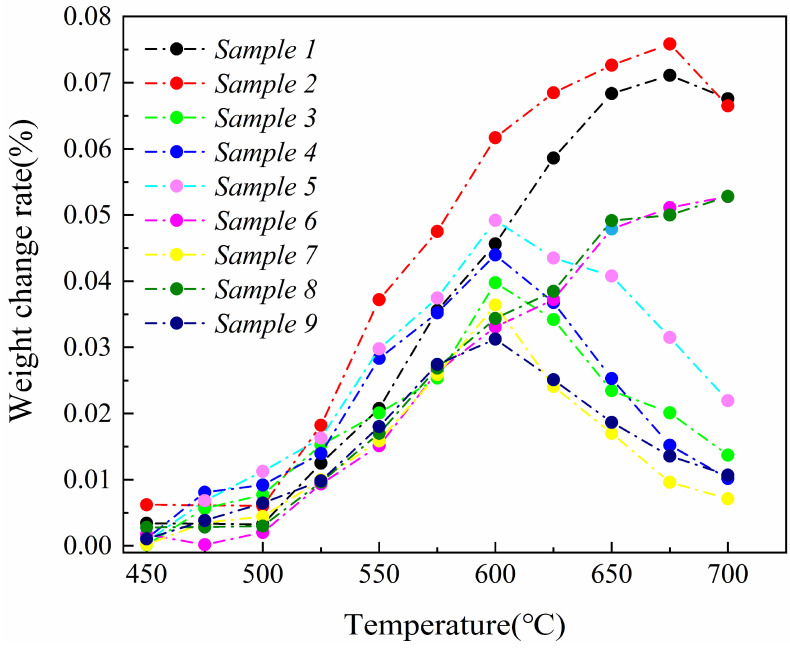
Comparison of weight change rate of the composite coating orthogonal experiment samples in the thermal shock experiment.

**Figure 4 materials-17-05798-f004:**
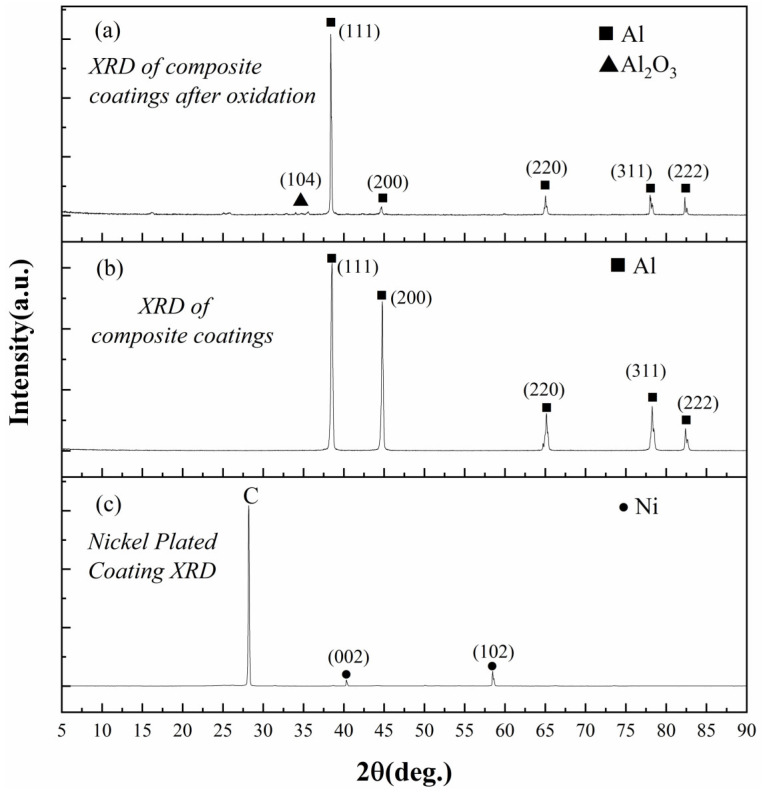
X-ray diffraction (XRD) pattern after coating and high-temperature oxidation. (**a**) XRD of composite coatings after oxidation, (**b**) XRD of composite coatings, and (**c**) XRD of nickel-plated coating.

**Figure 5 materials-17-05798-f005:**
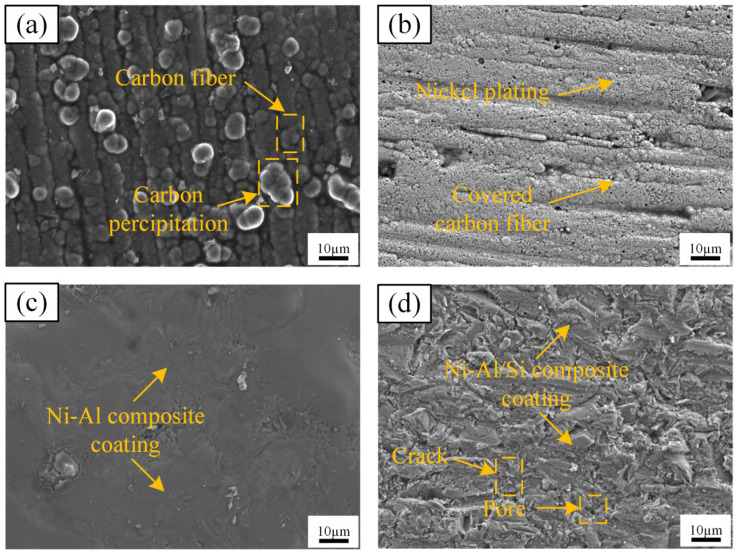
SEM image of coating surface: (**a**) C/C composite substrate, (**b**) Ni coating, (**c**) Ni-Al composite coating, and (**d**) Ni-Al/Si composite coating.

**Figure 6 materials-17-05798-f006:**
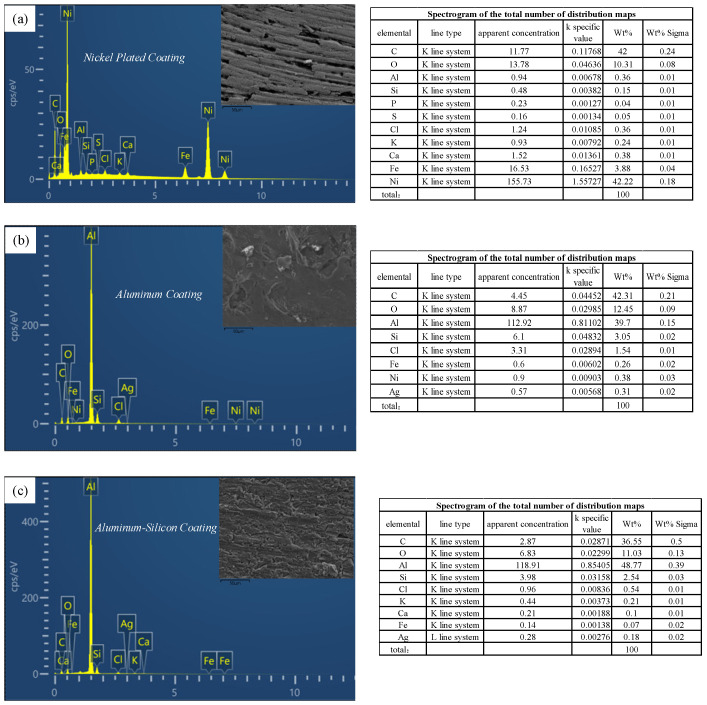
Surface energy spectrum of coatings: (**a**) Ni coating, (**b**) hot-dip aluminum coating, and (**c**) hot-dip Al/Si coating.

**Figure 7 materials-17-05798-f007:**
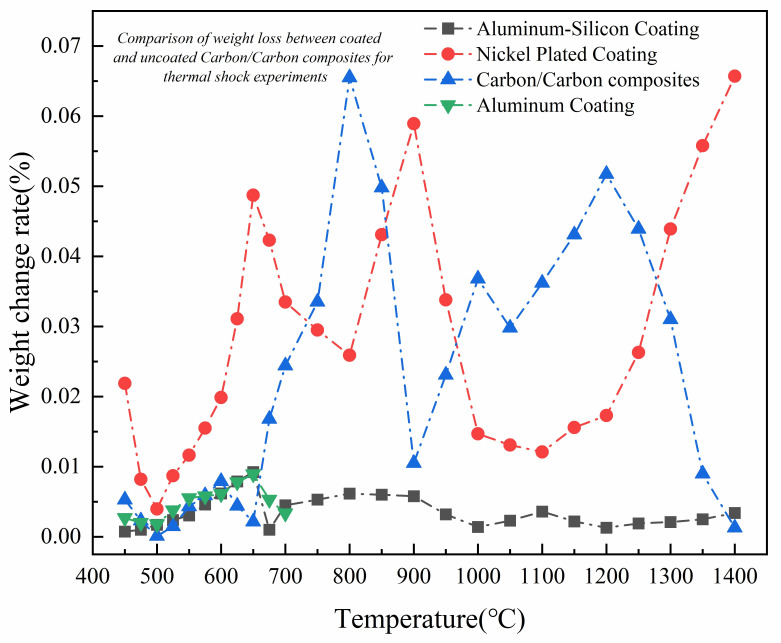
High-temperature oxidation weight change curve of coating and substrate.

**Figure 8 materials-17-05798-f008:**
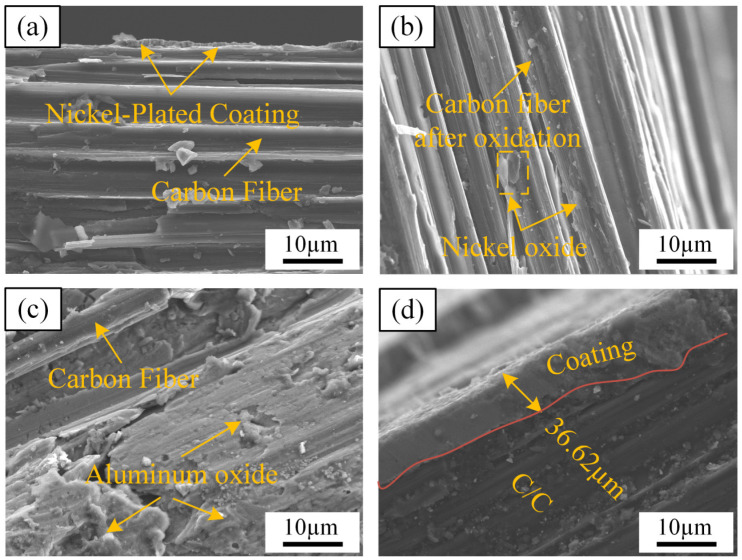
SEM morphology of oxidized coating cross-section: (**a**) electroplated Ni without oxidation, (**b**) electroplated Ni after oxidation, (**c**) hot-dip aluminum after oxidation, and (**d**) thickness of aluminum coating after hot-dip plating.

**Figure 9 materials-17-05798-f009:**
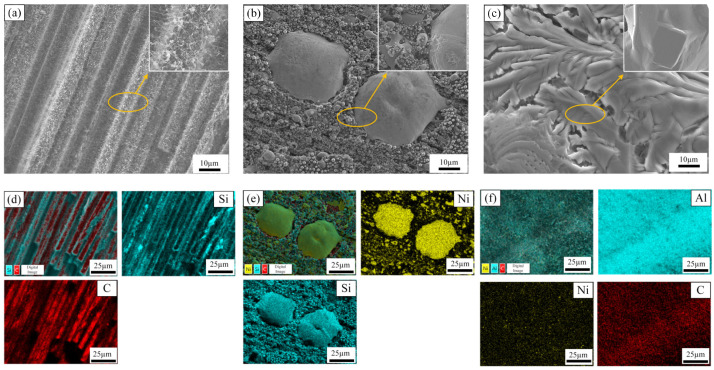
Scanning electron micrographs and elemental distribution of specimens after silicon vapor corrosion: (**a**) C/C composites after silicon corrosion. (**b**) Electroplated Ni specimens after silicon corrosion. (**c**) Hot-dip aluminum specimen after corrosion. (**d**) Elemental distribution of C/C composites after corrosion. (**e**) Elemental distribution of electroplated Ni specimens after corrosion. (**f**) Elemental distribution of hot-dip-plated specimens after silicon corrosion.

**Table 1 materials-17-05798-t001:** Electroplating solution formulation.

NiCO_3_	NiCl_2_	H_3_BO_3_	pH	Stirring
406 ${\mathrm{gL}}^{-1}$	45 ${\mathrm{gL}}^{-1}$	175.5 ${\mathrm{gL}}^{-1}$	5.8~6.2	500 rpm

NiCO_3_ (Shanghai Macklin Biochemical Technology Co., Ltd.) and NiCl_2_ (Shanghai Macklin Biochemical Technology Co., Ltd.) serve as sources of Ni ions for the solution, while boric acid acts as a buffer to regulate the pH of the electrolyte during the plating process. In the electrochemical reaction, variations in the pH of the solution influence the reaction rate at the electrode surface, the formation and precipitation of reaction products, and the protection of the electrode.

**Table 2 materials-17-05798-t002:** Composition of hot-dip plating aids.

Reagent	NaCl	KCl	NaF	KF
Mass Fraction	3.6%	30%	2.4%	2.4%

**Table 3 materials-17-05798-t003:** Factor level table for orthogonal experiments.

	Factor	A/min	B/°C	C/Silicon Content (wt %)
Level				
1		60	720	0
2		120	740	2
3		180	760	5

**Table 4 materials-17-05798-t004:** Orthogonal experimental data and computational analysis.

Experiment Number	A/min	B/°C	C/Silicon Content (%)	y/Oxidative Weight Change (%)
1	1	1	1	13.5108
2	1	2	2	17.2460
3	1	3	3	10.0913
4	2	1	2	11.2999
5	2	2	3	15.2080
6	2	3	1	9.6719
7	3	1	3	7.8758
8	3	2	1	10.2643
9	3	3	2	8.3390
I_1j_	40.8481	32.6865	33.4470	
II_2j_	36.1798	42.7183	36.8849	
III_3j_	26.4791	28.1022	33.1751	
K_1j_	13.6160	10.8955	11.1490	
K_2j_	12.0599	14.2394	12.2950	
K_3j_	8.8264	9.3674	11.0584	
R	4.7896	4.8720	1.2366	

## Data Availability

The original contributions presented in the study are included in the article, further inquiries can be directed to the corresponding author.

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
