# Peer review of "Antioxidant Behavior of Carbon/Carbon Composites with Hot Dip Plating and Electroplating for Single-Crystal Furnaces"

_materials, 2024, doi:10.3390/ma17235798_

Round 1

Reviewer 1 Report

Comments and Suggestions for Authors

The paper titled "Antioxidant Behavior of Carbon/Carbon Composites with Hot Dip Plating and Electroplating for Single-Crystal Furnaces" meets the established criteria, as it clearly outlines the objective and scope of the study, employing predominantly clear and direct language. However, it is recommended that the text be further simplified to enhance comprehension for a broader audience.             

The introductory section provides a solid context by situating the topic of silicon solar cells and the utilisation of Carbon/Carbon (C/C) composites within both past and current literature. Relevant research, such as that by Song and Yu (2021), is referenced to justify the significance of these materials in the context of efficiency and cost reduction in solar cell production. Furthermore, the increasing demand for thermal stability and quality in single-crystal furnaces is highlighted, reinforcing the study's relevance in a continually evolving field.

A careful selection of significant literature in the area of C/C composites and their application in the production of single-crystal furnaces is evident. Citations from previous studies, such as those by Fu et al. (2022) and Han et al. (2023), demonstrate an effort to include works that address both the properties of the composites and their limitations. However, clarity could be improved by articulating the specific criteria used for selecting these references, which would assist readers in better understanding the foundation upon which the argument is constructed.

Despite the presentation of relevant background information, the necessity of the article could be emphasised more explicitly. While the limitations regarding the oxidation resistance of C/C composites are mentioned, further elaboration on why this study is essential compared to previous research would be beneficial. Including an analysis that clearly articulates how the proposed novel approach to the preparation of innovative coatings not only addresses these limitations but also contributes to the advancement of the field would underscore the article's relevance in the current context.

I recommend reviewing the flow of this section to enhance clarity and conciseness. For instance, instead of:
"The detailed experimental procedure is comprehensively illustrated in Figure 2, providing a thorough representation of the steps involved,"
it could be reformulated to:
"The detailed experimental procedure is illustrated in Figure 2, which comprehensively represents the steps involved."

To improve clarity in this section, consider rephrasing the statement:
"The energy dispersive spectrometer (EDS) is employed for the quantitative analysis of the elemental composition in the microregions of the coating,"
to a more simplified version, such as:
"An energy dispersive spectrometer (EDS) is used for the quantitative analysis of the elemental composition in the microregions of the coating."

It is advisable to condense the information for greater effectiveness. Instead of:
"In this experiment, a box-type resistance furnace is used to perform thermal shock tests at temperatures of 450℃, 500℃, 550℃, 600℃, 650℃, and 700℃,"
the alternative could be:
"Thermal shock tests are conducted in a box-type resistance furnace at temperatures of 450℃, 500℃, 550℃, 600℃, 650℃, and 700℃."

Given that the silicon content had a relatively minor impact on the antioxidant properties, would it be advisable to utilise silicon levels at the minimum range necessary to avoid complications in the coating process?

Could it be that the issue lies in the insufficient polishing of the C/C substrates prior to the preparation of the coating, which would affect both adhesion and uniformity? This may minimise the occurrence of surface imperfections that could compromise performance.

Is it feasible to implement continuous temperature monitoring during thermal shock experiments, particularly beyond 600 °C, to observe the effect of the formation of the protective Al₂O₃ layer on the coating's performance?

Furthermore, pre-oxidation processes on Al substrates could be considered to enhance the formation of the protective layer and increase oxidation resistance.

Lastly, it would be prudent to conduct additional analyses using XRD and SEM at various time intervals following the oxidative treatment, to monitor the evolution of the coating phases and the potential formation of new structures that could affect oxidation resistance.

Author Response

For research article

Response to Reviewer 1 Comments

Thank you for your letter and for the reviewers’ comments concerning our manuscript entitled “Antioxidant Behavior of Carbon/Carbon Composites with Hot Dip Plating and Electroplating for Single-Crystal Furnaces” with ID. materials-3299144. Those comments are all valuable and very helpful for revising and improving our paper, as well as the important guiding significance to our researches. We have studied comments carefully and have made correction which we hope meet with approval. Revised portion are marked in red and yellow in the paper. The main corrections in the paper and the responses to the reviewer’s comments are as following.

Comments 1: [The paper titled "Antioxidant Behavior of Carbon/Carbon Composites with Hot Dip Plating and Electroplating for Single-Crystal Furnaces" meets the established criteria, as it clearly outlines the objective and scope of the study, employing predominantly clear and direct language. However, it is recommended that the text be further simplified to enhance comprehension for a broader audience.]

Response 1: [We have simplified the text to improve the reader's comprehension.] Thank you for pointing this out. We agree with this comment. Therefore, We have taken your example and made modifications to it ["The detailed experimental procedure is comprehensively illustrated in Figure 2, providing a thorough representation of the steps involved," has been corrected to "The detailed experimental procedure is illustrated in Figure 2, which comprehensively represents the steps involved." – page 4, paragraph 2, and line 143-148.

"The energy dispersive spectrometer (EDS) is employed for the quantitative analysis of the elemental composition in the microregions of the coating," has been corrected to "An energy dispersive spectrometer (EDS) is used for the quantitative analysis of the elemental composition in the microregions of the coating." – page 5, paragraph 3, and line197

"In this experiment, a box-type resistance furnace is used to perform thermal shock tests at temperatures of 450, 500, 550, 600, 650, and 700," has been corrected to "Thermal shock tests are conducted in a box-type resistance furnace at temperatures of 450, 500, 550, 600, 650, and 700." – page 5, paragraph 4, and line 206-207]

Comments 2: [The introductory section provides a solid context by situating the topic of silicon solar cells and the utilisation of Carbon/Carbon (C/C) composites within both past and current literature. Relevant research, such as that by Song and Yu (2021), is referenced to justify the significance of these materials in the context of efficiency and cost reduction in solar cell production. Furthermore, the increasing demand for thermal stability and quality in single-crystal furnaces is highlighted, reinforcing the study's relevance in a continually evolving field.]

Response 2: We revised manuscript to emphasize this point. [Fu et al. consolidate recent advancements in micro/nano multiscale strategies—such as nanoparticles, nanowires, and graphene—to enhance high-temperature oxidation and ablation resistance in C/C composites. It concludes with challenges and future directions for achieving superior mechanical and thermal performance, aiming to inspire scientific and industrial advancements in robust C/C materials. Han et al. investigate the impact of oxidation on residual mechanical properties for safe in-service use. They propose a microscale degradation model based on mass conservation and diffusion to predict oxidation behavior and use simulations to show a 24–32% strength reduction after 30 minutes at 850 °C. Their findings reveal that oxidation redis-tributes stress, weakens load capacity, and increases oxygen diffusion in stressed areas of the pyrocarbon matrix.

 Mention exactly where in the revised manuscript this change can be found – page 1,2, paragraph1, and line 41-45,48-53.]

Comments 3: [Given that the silicon content had a relatively minor impact on the antioxidant properties, would it be advisable to utilise silicon levels at the minimum range necessary to avoid complications in the coating process?]

Response 3: Agree. Thank you for your insightful question. We agree that minimizing silicon content could reduce complications in the coating process, especially given its relatively minor impact on antioxidant properties. However, our C/C coating has been applied in industrial production, effectively addressing the issue of silicon vapor corrosion in Czochralski single crystal silicon furnaces. The minimum silicon content may not be suitable for practical industrial applications.

Comments 4: [Could it be that the issue lies in the insufficient polishing of the C/C substrates prior to the preparation of the coating, which would affect both adhesion and uniformity? This may minimise the occurrence of surface imperfections that could compromise performance.]

Response 4: Agree. Thank you for the valuable suggestion. We acknowledge that substrate preparation, including sufficient polishing, plays a crucial role in coating adhesion and uniformity. In our study, we carefully polished the C/C substrates to ensure a smooth surface prior to coating application. We agree that further optimizing the polishing process may enhance coating performance by reducing surface imperfections.

Comments 5: [Is it feasible to implement continuous temperature monitoring during thermal shock experiments, particularly beyond 600 °C, to observe the effect of the formation of the protective Al₂O₃ layer on the coating's performance?]

Response 5: Agree. Thank you for this insightful suggestion. Continuous temperature monitoring during thermal shock experiments beyond 600 °C would indeed provide valuable data on the formation and effect of the Al₂O₃ protective layer. While real-time monitoring can be challenging due to the high temperatures involved, we recognize its potential to enhance understanding of coating performance. We will consider incorporating this approach in future experiments and discuss the potential impact of the Al₂O₃ layer formation on thermal shock resistance in the revised manuscript.

Comments 6: [Furthermore, pre-oxidation processes on Al substrates could be considered to enhance the formation of the protective layer and increase oxidation resistance].

Response 6: Agree. Thank you for your thoughtful suggestion regarding the pre-oxidation of Al substrates. We agree that a pre-oxidation process could promote the formation of a more stable Al₂O₃ protective layer, potentially enhancing oxidation resistance. This approach will be considered in future work to optimize coating performance.

Comments 7: [Lastly, it would be prudent to conduct additional analyses using XRD and SEM at various time intervals following the oxidative treatment, to monitor the evolution of the coating phases and the potential formation of new structures that could affect oxidation resistance.]

Response 7: Agree. Thank you for this valuable recommendation. We agree that conducting additional XRD and SEM analyses at different time intervals post-oxidative treatment would provide important insights into the phase evolution and potential new structure formation within the coating, which could influence its oxidation resistance. We will consider incorporating these analyses in future studies.

Thanks again for your comments

Reviewer 2 Report

Comments and Suggestions for Authors

MANUSCRIPT REVIEW

Manuscript title: Antioxidant Behavior of Carbon/Carbon Composites with Hot Dip Plating and Electroplating for Single-Crystal Furnaces

Reviewer’s Comments:

1.      Please, separate the names of first and second authors with commas as in the template; it is confusing written like this.

2.      Please use the same way of writing the name of method, in the title is defined differently (Hot Dip Plating) and then hot-dip plating in the abstract, and then Hot dipping in key words.

3.      Please, read instruction for authors and use template for this Journal. References are not listed at the journal's request, see how they are listed. “In the text, reference numbers should be placed in square brackets [ ], and placed before the punctuation; for example [1], [1–3] or [1,3]. For embedded citations in the text with pagination, use both parentheses and brackets to indicate the reference number and page numbers; for example [5] (p. 10). or [6] (pp. 101–105).” References are not edited according to the instructions for authors.

4.      The introduction part contains a good review of the literature and recent references are included. It seems a bit disjointed. The problem is that the goal of this research is not indicated anywhere in the Introduction part. Write a summary of the aim and contribution of your research.

5.      In the text, please use Figure X, or Fig. X, establish a marking method.

6.      Lines 151-154 and lines 154-157, are the same sentences, delete them.

7.      Lines 161-163 and lines 163-165, repeating again, no need.

8.      Table 1 Electroplating solution formulation, you have already stated in the description of the experiment the conditions, accurately define the type of mixing (rpm?), then you can transfer the salt concentrations to where you first mentioned them.

9.      Please, define the layer thicknesses after electroplating and hot-dipping methods.

10.  Table 4, “Actor Level Experiment number”, typo-error, please correct.

11.  Figure 3- Try to correct the font and make it uniform on all graphics. This looks bad.

12.  Figure 4- Please, state the characteristic crystallographic orientations, their corresponding angles and preferably cite adequate literature or standards for diffraction peaks. The proposal is to extract crystallinity and grain size.

13.  Line 299-300, are the same parts of sentences, deleted.

14.  Figure 5(c) is not Ni, it is Al or not? The authors wrote “In Fig. 5(b) and 5(c), the Ni coating exhibits a few surface holes, but overall coverage is adequate.” What is the cause of the appearance of the holes in Figure 5(b)?

15.  The EDS analyses- please emphasize in the text whether the percentages are wt. % or atomic %. There are differences.

16.  Lines 341-343, the authors wrote: “The presence of Si contributes to increased hardness and wear resistance of the Al/Si coatings compared to the other two coatings, although it also makes the preparation process more challenging.” How did you make this claim? The paper does not deal with the examination of the mechanical properties of the coatings, a reference is required.

17.  For thermal shock experiment, are not the most clearly connected graphs in Figure 3 and Figure 7? Designing an optimal experiment (DoE) is one thing, confirmation by experiment is another. What is the correlation?

18.  Line 380-382, the authors wrote: “According to the experimental results, the prepared coatings exhibit excellent adhesion to the substrate materials and effectively enhance their antioxidant properties”. How did you assess adhesion, based on what? Delamination process it can also be a consequence of non-adequate preparation of the cross-section. In the case you do not perform metallographic preparation of the cross-section; you have the growth of the Ni layer on the side of the sample. How can you be sure that the oxides are on the cross-section, when no elemental analysis of the cross-section of the samples has been done?

19.  Figure 9 is excellently done; please try to comment the details a little deeper.

20.  The practical, an academic, significance, and industrial applications, only mentioned, but clearly stated examples are not suggested. State where synthesized composites can be applied.

October 29 2024

Author Response

For research article

Response to Reviewer 2 Comments

Thank you for your letter and for the reviewers’ comments concerning our manuscript entitled “Antioxidant Behavior of Carbon/Carbon Composites with Hot Dip Plating and Electroplating for Single-Crystal Furnaces” with ID. materials-3299144. Those comments are all valuable and very helpful for revising and improving our paper, as well as the important guiding significance to our researches. We have studied comments carefully and have made correction which we hope meet with approval. Revised portion are marked in red and yellow in the paper. The main corrections in the paper and the responses to the reviewer’s comments are as following.

Comments 1: [Please, separate the names of first and second authors with commas as in the template; it is confusing written like this.]

Response 1: Thank you for pointing this out. We agree with this comment. Therefore, We have revised authors. [Zuxing Qi a, Chaofan Du a**, Guoying Bao a, Shan Wang a*, Dedong Gao a, Haixing Lin a, Yan An b, – page 1, paragraph 1, and line 4.]

Comments 2: [Please use the same way of writing the name of method, in the title is defined differently (Hot Dip Plating) and then hot-dip plating in the abstract, and then Hot dipping in key words.]

Response 2: We have modified manuscript to emphasize this point. We have standardised hot dip plating throughout the text. It's consistent with the title.

Comments 3: [Please, read instruction for authors and use template for this Journal. References are not listed at the journal's request, see how they are listed. “In the text, reference numbers should be placed in square brackets [ ], and placed before the punctuation; for example [1], [1–3] or [1,3]. For embedded citations in the text with pagination, use both parentheses and brackets to indicate the reference number and page numbers; for example [5] (p. 10). or [6] (pp. 101–105).” References are not edited according to the instructions for authors.]

Response 3: We have modified manuscript to emphasize this point. For the citation of references, we have modified it to the MDPI manuscript format

Comments 4: [The introduction part contains a good review of the literature and recent references are included. It seems a bit disjointed. The problem is that the goal of this research is not indicated anywhere in the Introduction part. Write a summary of the aim and contribution of your research.]

Response 4: We have supplemented manuscript to emphasize this point. We have revised the content of the manuscript and added relevant aims and contributions. [Considering the challenges associated with depositing metal coatings on C/C com-posites, which arise from their low thermal expansion coefficient and intricate carbon fi-ber structure. A novel method is proposed for the preparation of Ni-Al and Ni-Al/Si com-posites coatings on C/C composites substrates, involving Ni electroplating followed by Al hot dip plating. This method reduces corrosion of the boiler by silicon vapour in the Czo-chralski single crystal silicon manufacturing industry. – page 4, paragraph 2, and line 143-148.]

Comments 5: [In the text, please use Figure X, or Fig. X, establish a marking method.]

Response 5: Thank you for pointing this out. Throughout the text we have uniformly modified Fig. X.

Comments 6: [Lines 151-154 and lines 154-157, are the same sentences, delete them.]

Response 6: Thank you for pointing this out. We sincerely apologise for this oversight. We've removed the repetition.

Comments 7: [Lines 161-163 and lines 163-165, repeating again, no need.]

Response 7: Thank you for pointing this out. We sincerely apologise again for this oversight. We've removed the repetition.

Comments 8: [Table 1 Electroplating solution formulation, you have already stated in the description of the experiment the conditions, accurately define the type of mixing (rpm?), then you can transfer the salt concentrations to where you first mentioned them.]

Response 8: Agree. Thank you for pointing this out. We have transfered the salt concentration within the first mentioned. – page 4, paragraph 6, and line 168.

Comments 9: [Please, define the layer thicknesses after electroplating and hot-dipping methods.]

Response 9: Thank you for pointing this out. We have defined the layer thicknesses after electroplating and hot-dipping methods. [In Fig. 8(d), the layer thickness after the electroplating and hot dipping methods is 36.62 μm – page 12, paragraph 2, and line 409,414.]

Comments 10: [Table 4, “Actor Level Experiment number”, typo-error, please correct.]

Response 10: Thank you for pointing this out. We sincerely apologise again for this oversight. We've corrected the typo-error. [Table 4, “Experiment number”, – page 7, paragraph 1, and line253.] Details of the numbering in the table are in the Appendix A. – page 15, paragraph 3-5, and line 473-479.

Comments 11: [Figure 3- Try to correct the font and make it uniform on all graphics. This looks bad.]

Response 11: Thank you for pointing this out. We sincerely apologise again for this oversight. We've corrected the Figure. – page 8,12 paragraph 1, and line 280,394.

Fig. 3 Comparison of weight change rate of composite coating orthogonal experiment samples for thermal shock experiment

Fig. 7 High temperature oxidation weight change curve of coating and substrate

Comments 12: [Figure 4- Please, state the characteristic crystallographic orientations, their corresponding angles and preferably cite adequate literature or standards for diffraction peaks. The proposal is to extract crystallinity and grain size.]

Response 12: Agree. Thank you for pointing this out. We have modified manuscript to emphasize this point. [After comparison with the PDF(98-000-0062) standard card, in Fig. 4(a), the peaks of Al are mainly 38.474°(111),44.723°(200), 65.1°(220), 78.234°(311), 82.441°(222). After compar-ison with the PDF(04-002-3621) standard card, in Fig. 4(a), the peaks of Al2O3 is 35.133°(104). After comparison with the PDF(98-000-0062) standard card, in Fig. 4(b), the peaks of Al are mainly 38.474°(111),44.723°(200), 65.1°(220), 78.234°(311), 82.441°(222). After com-parison with the PDF(04-002-8298) standard card, in Fig. 4(c), the peaks of Ni are 41.785°(002) and 58.532°(102) – page 8, paragraph 1, and line 284-290. ]

Comments 13: [Line 299-300, are the same parts of sentences, deleted.]

Response 13: Agree. Thank you for pointing this out. We've deleted the same parts. – page 9, paragraph 1, and line 315.

Comments 14: [Figure 5(c) is not Ni, it is Al or not? The authors wrote “In Fig. 5(b) and 5(c), the Ni coating exhibits a few surface holes, but overall coverage is adequate.” What is the cause of the appearance of the holes in Figure 5(b)?]

Response 14: Agree. Thank you for pointing this out. We have revised Figure 5 to emphasize this point. The oxidisation process has created holes in the coating but this does not affect the overall encapsulation. [The C/C composites can produce hole defects during oxidation due to thermal exposure area and oxidation time [36]. Adsorption on the surface of the substrate despite the pres-ence of defects– page 9, paragraph 1, and line 317-319.]

Comments 15: [The EDS analyses- please emphasize in the text whether the percentages are wt. % or atomic %. There are differences.]

Response 15: Agree. Thank you for pointing this out. In our EDS analyses we use weight percentages.

Comments 16: [Lines 341-343, the authors wrote: “The presence of Si contributes to increased hardness and wear resistance of the Al/Si coatings compared to the other two coatings, although it also makes the preparation process more challenging.” How did you make this claim? The paper does not deal with the examination of the mechanical properties of the coatings, a reference is required.]

Response 16: Agree. Thank you for pointing this out. We have added references [37]. It is mentioned in this literature: The increase in ultimate tensile strength with increasing Si content up to that giving a completely eutectic microstructure is explained by a redistribution of volume content of α-Al and eutectic. The increase in tensile strength with increasing rate is explained by a decrease in microstructural scale accompanying the transformation of flake-to-fiber eutectic microstructure. The optimal fine fiber structure without any primary crystals of Al-Si alloy at a given Si content is obtained at the solidification rate giving a completely eutectic microstructure at that composition.

Comments 17: [For thermal shock experiment, are not the most clearly connected graphs in Figure 3 and Figure 7? Designing an optimal experiment (DoE) is one thing, confirmation by experiment is another. What is the correlation?]

Response 17: Agree. Thank you for pointing this out. The use of samples with slightly weaker antioxidant properties exposes possible defects in the oxidation process more than the optimal solution and is more valuable for analysis.

Comments 18: [Line 380-382, the authors wrote: “According to the experimental results, the prepared coatings exhibit excellent adhesion to the substrate materials and effectively enhance their antioxidant properties”. How did you assess adhesion, based on what? Delamination process it can also be a consequence of non-adequate preparation of the cross-section. In the case you do not perform metallographic preparation of the cross-section; you have the growth of the Ni layer on the side of the sample. How can you be sure that the oxides are on the cross-section, when no elemental analysis of the cross-section of the samples has been done?]

Response 18: Agree. Thank you for pointing this out. We revised the manuscript. we ensured proper sample preparation to minimize delamination issues that could arise from inadequate cross-sectional preparation. For the test of coating adhesion is a more complex process, we embodied the process of sticking during the experiment. Therefore we have deleted the excellent adhesion properties from the manuscript. The identification of the oxides was derived from a previously published paper [DOI:10.16577/j.issn.1001-1560.2023.0171]. [As shown in Fig. 8, the coating adheres to the the substrate materials. The SEM observa-tions indicate that the unoxidized electroplated Ni coatings in Fig. 8 (a) exhibit a laminat-ed structure. This structure provides effective protection to the substrate materials– page 12, paragraph 2, and line 398-401.]

Comments 19: [Figure 9 is excellently done; please try to comment the details a little deeper.]

Response 19: Thank you for pointing this out. We've added some descriptive details. [Ultimate tensile strength increases with increasing Si content in Al/Si alloys [37] – page 13, paragraph 1, and line 423-424.

In the EDS analysis, Si is not tested in the mapping pattern due to its low content– page 13, paragraph 1, and line 430-431.]

Comments 20: [The practical, an academic, significance, and industrial applications, only mentioned, but clearly stated examples are not suggested. State where synthesized composites can be applied.]

Response 20: Thank you for pointing this out. Our research is based on monocrystalline silicon companies. We solve the problem of silicon vapour corrosion in real-life production for companies. The specific manufacturer is Sichuan Gokin Solar Technology Co.

Thanks again for your comments

Reviewer 3 Report

Comments and Suggestions for Authors

In my opinion, the article is written a bit chaotic and has some shortcomings. Literature references are relevant and up-to-date.

I suggest making some corrections:

1. In the 2.2. Coating Characterization section, some of the research methods should be described in more detail. The description and working parameters of the scanning electron microscope applied should be given, i.e., acceleration voltage, work distance, the kind of images (SEI or BEI). Similarly, for EDS, more details should be provided. The type of anode in case of XRD should be given. In addition, the type software should be specified.

2. How were the process parameters determined in the orthogonal experiment - based on previous studies or literature? This should be clearly stated in the text

3. Lines 151-156: There is a repetition here.

4. Line 223 - the range R calculation method - what did the calculations consist of, what is the procedure, was any software used?

5. Table 4 contains symbols I, II, III, K1, K2, K3, R, which are not described in any way in the text. This should be supplemented.

6. Why in Table 4 the results "y/oxidative weight loss (%)" are in the order of a few or a dozen, while in Figure 3 the results "weight loss rate (%)" are in the order of a few hundredths.

7. Figure 4 - What specific coatings do these diffractograms refer to? The caption under this drawing is also unclear.

8. Figure 5 c and d - What specific coatings do these images refer to?

9. Throughout the text the authors use the phrase "weight loss" when it should rather be "weight change". In each case, no weight loss is observed, only weight gain.

10. In chapter 3.5. Coating high temperature oxidation performance, the authors describe what can be seen in graph 7. Unfortunately, the authors do not even attempt to explain what is shown in graph 7. There is no scientific explanation of the phenomena here. If this is a scientific article, it should be supplemented.

11. Silicon mapping should be included in Figure 9f.

12. The authors wrote in line 346 what the high content of C and O results from, but they did not take into account the fact that these elements are imprecisely estimated by the EDS method. If they had used the WDS method, the amount of oxygen and carbon would be precise. Otherwise, drawing conclusions based on the EDS method is inappropriate.

Author Response

For research article

Response to Reviewer 3 Comments

Thank you for your letter and for the reviewers’ comments concerning our manuscript entitled “Antioxidant Behavior of Carbon/Carbon Composites with Hot Dip Plating and Electroplating for Single-Crystal Furnaces” with ID. materials-3299144. Those comments are all valuable and very helpful for revising and improving our paper, as well as the important guiding significance to our researches. We have studied comments carefully and have made correction which we hope meet with approval. Revised portion are marked in red and yellow in the paper. The main corrections in the paper and the responses to the reviewer’s comments are as following.

Comments 1: [In the 2.2. Coating Characterization section, some of the research methods should be described in more detail. The description and working parameters of the scanning electron microscope applied should be given, i.e., acceleration voltage, work distance, the kind of images (SEI or BEI). Similarly, for EDS, more details should be provided. The type of anode in case of XRD should be given. In addition, the type software should be specified.]

Response 1: Thank you for pointing this out. We agree with this comment. Therefore, we have revised manuscript. [The field FESEM offers high resolution, with secondary electron images reaching 0.7 nm at 15 kV. It has a magnification range from 25x to 1,000,000x and an acceleration voltage range of 0.01 kV to 30 kV. The microscope features a Schottky field emission electron gun with an electron beam current range of 1 pA to 500 nA. The working distance is automat-ically adjustable from 2 mm to 40 mm, with manual adjustment within 2 mm. The opti-mal working distance for EDS is 10 mm. The XRD (Rigaku) uses a copper (Cu) target as the anode, with a scintillation counter. Test mode settings include a step size of 0.02 degrees and a speed of 8 degrees per minute, using a 2-theta/theta linkage. Data analysis is conducted using JADE software.. – page 5, paragraph 3, and line191-198,200-203]

Comments 2: [How were the process parameters determined in the orthogonal experiment - based on previous studies or literature? This should be clearly stated in the text.]

Response 2: Agree. Thank you for pointing this out. We have modified previous literature to emphasize this point. [Chen et al. investigate Cr-coated Zr alloys as promising ATF cladding materials. In their study, they apply Cr coatings to 1400 mm N36 tubes using an industrial arc system. Or-thogonal analyses identify key process parameters affecting coating characteristics, in-cluding surface roughness, defects, and crystal orientation – page 2, paragraph 2, and line 87-90.]

Comments 3: [Lines 151-156: There is a repetition here.]

Response 3: Agree. Thank you for pointing this out. We sincerely apologise for this oversight. We've removed the repetition.

Comments 4: [Line 223 - the range R calculation method - what did the calculations consist of, what is the procedure, was any software used?]

Response 4: Agree. Thank you for pointing this out. We have modified Appendix A. to emphasize this point. We did not use statistical software to calculate. [R is the range, the difference between the maximum and the minimum values of I, II, and III– page 15, paragraph 3, and line 473-479.]

Comments 5: [Table 4 contains symbols I, II, III, K1, K2, K3, R, which are not described in any way in the text. This should be supplemented.]

Response 5: Agree. Thank you for pointing this out. We have supplemented Appendix A. to emphasize this point. [Table A shows the schema of range calculations in the L9(33) orthogonal experiments. In this table, the letters “A–C” represent the influencing factors investigated and the corresponding numbers “1–3” are the levels set for these factors. I1j, II2j, and III3j are the sums of the results corresponding to the m-th level of the factor in column j. kmj is the average value of I1j, II2j, and III3j. R is the range, the difference between the maximum and the minimum values of I, II, and III– page 15, paragraph 3, and line 473-479.]

Table A. Schema of range calculations in the L9(33) orthogonal experiments.

No.

A

B

C

Index

1

1

1

1

X1

2

1

2

2

X2

3

1

3

3

X3

4

2

1

2

X4

5

2

2

3

X5

6

2

3

1

X6

7

3

1

3

X7

8

3

2

1

X8

9

3

3

2

X9

I1j

I11= X1+ X2+ X3

I12= X1+ X4+ X7

I13= X1+ X6+ X8

II2j

II21= X4+ X5+ X6

II22= X2+ X5+ X8

II23= X2+ X4+ X9

III3j

III 31= X7+ X8+ X9

III 32= X3+ X6+ X9

III 33= X3+ X5+ X7

K1j

I11/3

I12/3

I13/3

K2j

II21/3

II22/3

II23/3

K3j

III 31/3

III 32/3

III 33/3

R

(Max-Min)

(I11, II21, III 31)

(Max-Min)

(I12, II22, III 32)

(Max-Min)

(I13, II23, III 33)

Comments 6: [Why in Table 4 the results "y/oxidative weight loss (%)" are in the order of a few or a dozen, while in Figure 3 the results "weight loss rate (%)" are in the order of a few hundredths.]

Response 6: Agree. Thank you for pointing this out. We use two different calculation methods. In Table 4 is the ratio of the last weight to the first weight. However, the Figure 3 shows the ratio of each weight loss rate to the previous one.

Comments 7: [What specific coatings do these diffractograms refer to? The caption under this drawing is also unclear.]

Response 7: Agree. Thank you for pointing this out. We have revised manuscript to emphasize this point. [(a) XRD of composite coatings after oxidation (b) XRD of composite coatings (c) Nickel Plated Coating XRD– page 9, paragraph 1, and line 307.]

Comments 8: [Figure 5 c and d - What specific coatings do these images refer to?]

Response 8: Agree. Thank you for pointing this out. We have revised Figure 5 to emphasize this point. [Figure. 5 SEM image of coating surface (a) C/C composites substrate (b) Ni coating (c) Ni-Al composites coating (d) Ni-Al/Si composites coating– page 9, paragraph 1, and line 317-319.]

Comments 9: [Throughout the text the authors use the phrase "weight loss" when it should rather be "weight change". In each case, no weight loss is observed, only weight gain.]

Response 9: Agree. Thank you for pointing this out. We have replaced ‘weight loss’ with ‘weight change’ throughout the full text.

Comments 10: [In chapter 3.5. Coating high temperature oxidation performance, the authors describe what can be seen in graph 7. Unfortunately, the authors do not even attempt to explain what is shown in graph 7. There is no scientific explanation of the phenomena here. If this is a scientific article, it should be supplemented.]

Response 10: Agree. Thank you for pointing this out. Therefore, we have revised manuscript. [This increase is attributed to the oxidizing reaction between the Ni in the coating and the air, with some of the oxidized material escaping from the specimen's surface, thereby in-creasing the weight change rate. During the oxidation process, the weight change rate de-creases due to the presence of Ni oxide – page 12, paragraph 1, and line 382-385.

This decrease is attributed to the formation of Al₂O₃ on the surface of the coatings, which inhibits further oxidation. This indicates that the composite coatings effectively enhance the antioxidant performance of C/C composites and exhibit better antioxidant properties compared to the individual Ni coatings. It is still able to maintain a low weight loss at 1400°C– page 12, paragraph 1, and line 389-393.]

Comments 11: [Silicon mapping should be included in Figure 9f.]

Response 11: Agree. Thank you for pointing this out. We measured Silicon during the experiment. However, due to the better coating encapsulation in the EDS no measurement of Silicon. [In the EDS analysis, Si is not tested in the mapping pattern due to its low content– page 13, paragraph 1, and line 430-431.]

Comments 12: [The authors wrote in line 346 what the high content of C and O results from, but they did not take into account the fact that these elements are imprecisely estimated by the EDS method. If they had used the WDS method, the amount of oxygen and carbon would be precise. Otherwise, drawing conclusions based on the EDS method is inappropriate.]

Response 12: Agree. Thank you for pointing this out. We use the WDS method. Therefore, the amounts of oxygen and carbon are precise.

Thanks again for your comments!

Round 2

Reviewer 2 Report

Comments and Suggestions for Authors

The authors answered all questions and clarified some unclearly written parts of the text.

Author Response

Thanks again for all your hard work!

Reviewer 3 Report

Comments and Suggestions for Authors

1. In response, the authors wrote:

“Comments 12: [The authors wrote in line 346 what the high content of C and O results from, but they did not take into account the fact that these elements are imprecisely estimated by the EDS method. If they had used the WDS method, the amount of oxygen and carbon would be precise. Otherwise, drawing conclusions based on the EDS method is inappropriate.] Response 12: Agree. Thank you for pointing this out. We use the WDS method. Therefore, the amounts of oxygen and carbon are precise.

This is not true, the authors did not use the WDS method. The results were not changed in any way.

The WDS method would be more precise here, but it is more complicated and time-consuming. There is no need for the authors to use this method, but they could at least comment better on the EDS results presented in the text of the manuscript.

For example, the authors in line 349 state the accuracy of the oxygen measurement to two decimal places, which is absurd for this element and this method.

I suggest the authors learn more about this EDS method and describe the results obtained with this method better.

2. In response, the authors wrote:

“Comments 9: [Throughout the text the authors use the phrase "weight loss" when it should rather be "weight change". In each case, no weight loss is observed, only weight gain.] Response 9: Agree. Thank you for pointing this out. We have replaced ‘weight loss’ with ‘weight change’ throughout the full text.

Unfortunately now it is even worse, because both the phrases "weight loss" and "weight change" are still in the text at the same time. Please check the entire text again, including the drawings.

Author Response

For research article

Response to Reviewer 3 Comments

Thank you for your letter and for the reviewers’ comments concerning our manuscript entitled “Antioxidant Behavior of Carbon/Carbon Composites with Hot Dip Plating and Electroplating for Single-Crystal Furnaces” with ID. materials-3299144. Those comments are all valuable and very helpful for revising and improving our paper, as well as the important guiding significance to our researches. We have studied comments carefully and have made correction which we hope meet with approval. Revised portion are marked in red and yellow in the paper. The main corrections in the paper and the responses to the reviewer’s comments are as following.

Comments 9: [“Comments 9: [Throughout the text the authors use the phrase "weight loss" when it should rather be "weight change". In each case, no weight loss is observed, only weight gain.] Response 9: Agree. Thank you for pointing this out. We have replaced ‘weight loss’ with ‘weight change’ throughout the full text.”

Unfortunately now it is even worse, because both the phrases "weight loss" and "weight change" are still in the text at the same time. Please check the entire text again, including the drawings.]

Response 9: Thank you for pointing this out. We agree with this comment. Therefore, we have revised manuscript. [Fig. 3 Comparison of weight change rate of composite coating orthogonal experiment samples for thermal shock experiment – page 8, paragraph1, and line 280.

It is still able to maintain a low weight change at 1400°C.- – page 12, paragraph1, and line 393

 Fig. 7 High temperature oxidation weight change curve of coating and substrate – page 12, paragraph1, and line 395.]

Comments 12: [This is not true, the authors did not use the WDS method. The results were not changed in any way.

The WDS method would be more precise here, but it is more complicated and time-consuming. There is no need for the authors to use this method, but they could at least comment better on the EDS results presented in the text of the manuscript.

For example, the authors in line 349 state the accuracy of the oxygen measurement to two decimal places, which is absurd for this element and this method.

I suggest the authors learn more about this EDS method and describe the results obtained with this method better.”]

Response 9: Thank you for pointing this out. We've learnt about EDS and WDS. EDS (Energy Dispersive Spectroscopy) and WDS (Wavelength Dispersive Spectroscopy) are two analytical techniques commonly used in electron microscopy to determine the elemental composition of a sample. Both are based on X-ray detection but differ significantly in their operating principles and applications.

EDS (Energy Dispersive Spectroscopy)

Principle: Measures the energy of characteristic X-rays emitted by elements in the sample. Each element has a unique X-ray energy fingerprint, allowing for identification.

Detector: Uses a semiconductor detector (often silicon drift detectors, SDDs).

Advantages:

Fast data acquisition.

Suitable for qualitative and semi-quantitative analysis.

Simultaneous detection of multiple elements.

Compact and cost-effective.

Limitations:

Lower energy resolution (~120-140 eV at Mn Kα).

Limited accuracy for trace element detection.

Potential overlaps in X-ray peaks for some elements.

WDS (Wavelength Dispersive Spectroscopy)

Principle: Measures the wavelength of characteristic X-rays using a diffraction grating or crystal, following Bragg's law.

Detector: Uses a spectrometer with a dispersive crystal and a gas-flow or solid-state detector.

Advantages:

High energy resolution (~10 eV or better).

Excellent for quantitative analysis and trace element detection.

Can resolve overlapping peaks that EDS cannot distinguish.

Limitations:

Slower data acquisition compared to EDS.

Can only measure one element at a time.

Larger and more expensive equipment.

EDS is preferred for quick, broad-spectrum analysis, while WDS is chosen when high precision or resolving power is needed, such as in complex materials with overlapping peaks. Both techniques are often complementary and used together in advanced material characterization. We have revised manuscript. [It is important to note that the contents of C and O elements are 42.2% and 12.5%, respec-tively, primarily originating from the C/C composites substrate and various introduced factors. The mass fraction of C is 36.6%, which is lower than the previous two experiments, while the mass fraction of Al is 48.8%, which is higher than the previous two experiments. – page 10, paragraph 2,3 and line 348-350,355-357].

Thanks again for your comments!
